# Anxiety and Depression as Potential Predictors for Shorter Time to Undergo Initial Surgical Treatment for Papillary Thyroid Cancer

**DOI:** 10.3390/cancers16030545

**Published:** 2024-01-26

**Authors:** Dragan Vujovic, Mathilda Alsen, Vikram Vasan, Eric Genden, Maaike van Gerwen

**Affiliations:** 1Department of Otolaryngology-Head and Neck Surgery, Icahn School of Medicine at Mount Sinai, New York, NY 10029, USA; dragan.vujovic@icahn.mssm.edu (D.V.); mathilda.alsen@mountsinai.org (M.A.); eric.genden@mountsinai.org (E.G.); 2Institute for Translational Epidemiology, Icahn School of Medicine at Mount Sinai, New York, NY 10029, USA

**Keywords:** thyroid cancer, disparities, treatment delays, surgery, mental health, anxiety, depression

## Abstract

**Simple Summary:**

Papillary thyroid cancer diagnosis is a significant source of worry to patients, despite its relatively good prognosis. Such significant distress may prompt patients to receive surgical treatment, even when active surveillance may be a safer alternative. Patients with pre-existing anxiety and depression may be particularly prone to opting for surgical treatment. We collected data on the time between fine-needle aspiration (FNA) diagnosis and surgical treatment of a cohort of patients seen at a large tertiary center in New York in 2018–2020 and investigated mental health status as potential predictors. We found that pre-existing anxiety and depression may lead to shorter times for surgical intervention for thyroid cancer. Our study adds to the limited literature on mental health predictors of thyroid cancer decision making.

**Abstract:**

(1) Background: A pre-existing psychiatric condition may impact decision making by patients and/or physicians following a thyroid cancer diagnosis, such as potentially electing surgery over active surveillance, thus shortening the time to cancer removal. This is the first study to investigate the association between pre-existing anxiety and/or depression and time to receive surgical treatment for thyroid cancer. (2) Methods: Retrospective data were collected from 652 surgical thyroid cancer patients at our institution from 2018 to 2020. We investigated the time between thyroid cancer diagnosis and surgery, comparing patients with pre-existing anxiety and/or depression to those without. (3) Results: Patients with anxiety, depression, and both anxiety and depression had a significantly shorter time between diagnosis and surgery (51.6, 57, and 57.4 days, respectively) compared to patients without (111.9 days) (*p* = 0.002, *p* = 0.004, *p* = 0.003, respectively). (4) Conclusions: Although little is known about the impact of pre-existing psychiatric conditions in the decision-making process for thyroid cancer surgery, this present study showed that anxiety and/or depression may lead to more immediate surgical interventions. Thus, psychiatric history may be an important factor for physicians to consider when counseling patients with thyroid cancer.

## 1. Introduction

The steady increase in the incidence of thyroid cancer over the past decades can be partially attributed to the enhanced detection of small thyroid cancers (≤2 cm), for which diagnosis has increased four-fold from 1983 to 2011 [1]. Many of these newly detected tumors might have been either low-risk or subclinical diseases, as mortality did not significantly increase during that time [1]. The current American Thyroid Association (ATA) guidelines recommend lobectomy for low-risk thyroid cancer, which is defined as intrathyroidal differentiated thyroid carcinoma with no extrathyroidal extension, vascular invasion, or metastases [2]. Additionally, there is emerging evidence suggesting that active surveillance is an appropriate management strategy for low-risk thyroid cancer. A recent review of seven retrospective cohort studies noted that adults with subcentimeter, low-risk, papillary thyroid cancer, who elected for active surveillance, had no significant change in all-cause or cancer-specific mortality or increased risk of distant metastasis or recurrence compared to patients undergoing immediate surgical treatment [3]. Another study demonstrated excellent tumor outcomes in 286 patients with <1.5 cm cancers who elected for active surveillance, with none developing disease-specific metastasis or disease-associated death during the study period of at least 12 months [4]. Yet, even though these less invasive management strategies (lobectomy/active surveillance) are available, total thyroidectomies are most often performed, with up to 80% of surgeries being total thyroidectomies in small ≤2 cm tumors [5]. In fact, 38.2% of patients electing to have surgery had ≤1 cm tumors, despite the very good prognosis of microcarcinomas [6].

Opting for more invasive surgical management may be associated with psychological distress following thyroid cancer diagnosis. This distress is comparable to the level of distress experienced by patients with head and neck squamous cell cancer [7]. A study of 118 newly diagnosed thyroid cancer patients found that 43.3% suffered from clinically significant distress following diagnosis, even though 73% of the patients had early-stage disease and 94% of the patients had well-differentiated cancers [8]. A common worry is that the cancer would “grow and spread” and would cause patients to lose their speech and voice [9]. This excessive worry and negative emotions may prompt patients to seek definitive surgical care [10], as active surveillance could induce those negative emotions [11]. This is particularly relevant given the rising prevalence of mental health condition in recent years in the United States [12]; in 2019, nearly 20% of adults suffered from mental health conditions [12,13]. Previous diagnoses of depression or anxiety have been associated with significantly higher distress following thyroid cancer diagnosis (*p* = 0.011) [8].

To date, no studies have investigated the impact of a history of psychological distress on thyroid cancer management, while a study on prostate cancer showed that anxiety was a significant predictor of surgical treatment receipt, despite risks such as impotence and urinary incontinence [14]. Furthermore, breast cancer patients with higher levels of anxiety were more likely to elect for contralateral prophylactic mastectomy despite limited evidence that CPM improves survival [11,15]. Such differences between treatment decision making can be quantified by measuring the interval between diagnosis and treatment (TTI). This outcome has commonly been used as a surrogate for disparities, such as racial disparities, with TTI potentially indicative of hesitancy for receiving treatment or more barriers to care [16,17]. Given the rising prevalence of psychological disorders combined with the potential psychological distress of a cancer diagnosis possibly impacting cancer management decisions, this study investigated the association between pre-existing anxiety or depression and thyroid cancer management as well as the time to undergo surgical treatment for thyroid cancer in patients with papillary thyroid cancer.

## 2. Materials and Methods

### 2.1. Study Population

A retrospective chart review was conducted to collect demographic, clinical, and psychosocial data from surgically treated and pathologically confirmed thyroid cancer patients seen in the Department of Otolaryngology—Head and Neck Surgery at Mount Sinai Hospital in 2018–2020 (*n* = 770). Patients were excluded if they had benign disease, were <18 years old, or had a history of thyroid cancer surgery (*n* = 74). We further limited our study population to patients with papillary thyroid cancer. Of note, we removed one patient since their FNA diagnosis date could not be determined. This resulted in a final study population of 652 patients. This study was approved by the Program for the Protection of Human Subjects (PPHS) of the Icahn School of Medicine at Mount Sinai (STUDY-19-00730).

### 2.2. Data Collection

The medical records of included patients were retrospectively reviewed and data relevant to our study were collected and securely stored using Research Electronic Data Capture software (REDCap, Vanderbilt University, Nashville, TN, USA). Data were collected on age, sex, race, insurance status, family history of thyroid cancer, number of comorbidities, smoking status, surgery type, neck dissection status, and body mass index (BMI) with cut-offs used for normal weight (<25 kg/m^2^), overweight (25–30 kg/m^2^) and obese (>30 kg/m^2^), as proposed by the Center for Disease Control and Prevention (CDC) [18]. Data on comorbidities were translated into a categorical variable based on the number of comorbidities (0, 1, 2 or more). Our primary predictor was a history of anxiety, depression, or both prior to thyroid cancer diagnosis. We selected these two mental health predictors in our analysis, as they are the most frequent mental health disorders in the United States [19], as well as the most frequent encountered in our dataset. This information was obtained from a review of provider notes in the electronic medical record. Our primary outcome of interest was the time to treatment. This was defined as the difference in days between date of diagnosis based on the pathological report of fine-needle aspiration and the date of surgical intervention. Demographic, clinical, and histopathological characteristics were reported following SEER guidelines on data masking; exact cell sizes were masked to prevent the identification of patients [20].

### 2.3. Statistical Analysis

Demographic and clinical characteristics were compared between those with a pre-existing diagnosis of anxiety and/or depression and those without such diagnoses using a two-sided Student’s *t*-test for continuous variables and chi^2^ tests for categorical variables. Adjusted analysis was performed using multivariable linear regression modeling comparing the number of days between thyroid cancer diagnosis and surgery between patients with pre-existing anxiety, depression, and both anxiety and depression to those without, while adjusting for age, sex, race, insurance, and BMI. All statistical analyses were performed using SAS 9.4 (SAS Institute Inc., Cary, NC, USA).

## 3. Results

### 3.1. Study Population

Of the 652 patients with papillary thyroid cancer, 92 (14.1%) had a history of past anxiety and/or depression). There was no significant difference in age and sex between the patients with documented anxiety and/or depression diagnoses and those without. There was a significant difference in the distribution of patient race in patients, with more White patients having pre-existing anxiety alone and both anxiety and depression compared to those without (*p* < 0.0001 and *p* = 0.003, respectively). (Table 1). Additionally, there was a significant difference in patient insurance between the anxiety and both depression and anxiety groups (*p* = 0.006 and *p* = 0.022, respectively). Additionally, there was a significant difference regarding the family history of thyroid cancer in the patients who had both pre-existing anxiety and depression (*p* = 0.043).

### 3.2. Time to Treatment

Patients with pre-existing anxiety, depression, and both anxiety and depression had a significantly shorter time between diagnosis and surgical treatment (51.6, 57, and 57.4 days, respectively) compared to patients without anxiety or depression (111.9 days) (*p* = 0.002, *p* = 0.004, *p* = 0.003, respectively) (Table 2).

Univariate linear regression showed no significant difference in time to obtain treatment when comparing a history of mood disorder or history of anxiety, depression, or both anxiety and depression to the group without these psychiatric conditions—β: −60.31 (*p*-value = 0.402), β: −54.90 (*p*-value = 0.462), and β: −54.50 (*p*-value = 0.399), respectively. These results remained similar after adjustment—β_adj_: −50.87 (*p*-value= 0.489), β_adj_: −37.76 (*p*-value= 0.633), and β_adj_: −51.29 (*p*-value= 0.443), respectively (Table 2).

## 4. Discussion

The results of this present study suggest that the potential association between pre-existing anxiety and/or depression and a shorter time to initial surgical treatment for papillary thyroid cancer is a novel finding, as previous studies exclusively focused on thyroid cancer patients whose mental health is affected by their cancer diagnosis, treatment, and oncologic progression.

Previous studies have shown that depression and anxiety are common among cancer patients, with prevalence of up to 37% [21,22]. Several studies noted that thyroid cancer patients have substantial unmet psychosocial needs that may lead to elevated rates of psychological symptoms and mental health conditions after a thyroid cancer diagnosis [8,23,24,25]. Additionally, while this current study focuses on PTC, a majority of these aforementioned studies on psychosocial needs of thyroid cancer analyzed patients with PTC’s umbrella term, differentiated thyroid cancer (DTC). DTC includes two histologic subtypes, papillary and follicular, with papillary as the most common histology. Even though thyroid cancer has an excellent prognosis, patients diagnosed with DTC still report wide-ranging issues concerning health-related quality of life. Nickel et al. and Rossi et al. reported that quality of life decline is more prevalent in patients with DTC who underwent thyroidectomies compared to hemithyroidectomies [25,26]. Hedman et al. and Dionisi-Visi reported that, among DTC survivors, anxiety may be a partially hidden yet common occurrence among patients even many years after the end of treatments due to the fear of recurrence and risk of developing other cancers [24,27]. Similarly, Noto et al. most recently found that DTC survivors experience significant psychological distress [28]. It is worth noting that this study focused on patients diagnosed with anxiety and/or depression before thyroid cancer diagnosis and treatment.

The impact of pre-existing anxiety and depression on DTC management, including time to treatment, has not yet been explored, while this topic has been investigated in other cancers (e.g., oral cancer, prostate cancer, and breast cancer). A prospective study assessing the impact of anxiety and depression during the time awaiting oral cancer treatment found that [29] a delay between diagnosis and the beginning of treatment for oral cancer promoted an increase in anxiety and depression among patients. In cases of prostate cancer, cancer-related anxiety may influence patients to choose surgical treatment over active surveillance [14]. In cases of breast cancer, patients with high anxiety scores at initial consultations were nine times as likely to undergo aggressive surgery compared with patients with low anxiety, thus impacting thyroid cancer management. Our sample size limited further investigation of potential underlying mechanisms.

Given the literature on other cancers, the impact of a history of anxiety and depression on treatment is important among patients with DTC while considering the intersection between patient goals and ATA guidelines for lower-risk thyroid cancers with regard to surgical treatment versus active surveillance. In the past, physicians have been reluctant to offer active surveillance, given the lack of robust evidence, standardized guidelines, and standardized protocols, but they agree that active surveillance has been historically underused [30,31]. More recently, active surveillance has been proposed as and shown to be an appropriate management strategy for patients with low-risk DTC, since the mortality rates, recurrence rates, and other outcomes are similarly positive to surgical treatment [3]. However, health care may not fully understand the impact of pre-existing anxiety and depression on patient desires in the management of their DTC. Latini et al. emphasized the need for providing psychosocial support to men with anxiety in early prostate cancer, where active surveillance is often the preferred option [14]. This support is crucial, as decisions to undergo treatment against guidelines and not based on clinical presentation and disease progression can put these patients at increased risk of potentially unnecessary complications. Hence, a similar approach may be of interest for DTC patients with anxiety and/or depression to properly counsel them regarding their choice for treatment or active surveillance. A recent study on DTC found that both patients undergoing immediate surgery versus patients electing active surveillance for low-risk PTC treatment had equally significantly less distress associated with their treatment decision at the 6-month and 1-year follow-up points [32]. These results suggest an approach emphasizing patient autonomy in treatment selection, especially if research points towards similar prognoses between tumor resection and active surveillance. Nevertheless, our results highlight the importance of identifying patients with a history of anxiety and/or depression early on who could benefit from personalized support, extra educational services to fill in any knowledge gaps, and who may potentially not be good candidates for active surveillance.

It is interesting how anxiety, depression, and both anxiety and depression were all associated with a shorter time to treatment, even though they are both very different psychiatric conditions. It is understandable that individuals with anxiety disorder may experience a greater worry about their thyroid malignancy and would elect to have surgery sooner. In contract, individuals with depression may have fatigue and loss of energy, and have actually been shown to have slowed decision making [33]. Thus, it is conceivable that they may have a delay in the decision of having thyroid cancer surgery. One possible reason for the similar results between the two groups is that anxiety and depression have a very high degree of comorbidity with each other [34]. Thus, it may be difficult to separate the two conditions in a truly accurate way.

This study has some strengths and limitations. With regard to limitations, first, the borderline statistical significance for the linear regression results may be explained by the skewed data points in the cohort of patients without pre-existing anxiety and/or depression, with a few outlier patients who opted for a prolonged period of active surveillance. It is, however, reasonable to assume that only a few patients proceeded with active surveillance, as our dataset consisted of patients seen by head and neck surgeons; thus, the majority of patients were surgical candidates. Next, this study was conducted retrospectively at a single institution, impacting its generalizability. Next, the collection of the variables, anxiety and depression, were binary, lacking quantitative data about scores and levels at initial diagnosis. Additionally, while we restricted our cohort to papillary thyroid cancer, the least aggressive thyroid malignancy, there are other factors that may influence decision making by the physician, such as tumor size. Furthermore, while we adjusted for many social determinants of health, such as race/ethnicity and insurance, there are other potential confounders for treatment delays, such as health literacy or access to care. Finally, our sample size is limited, and more extensive studies with larger sample sizes are needed. The size of our sample also prevented us from further stratifying according to variables such as insurance type. For example, in our study, we found that a greater percentage of patients with both anxiety and depression have Medicare (30.6%) compared with our reference group of patients without any such mood disorder (11.8%). Thus, it is conceivable that patients on Medicare may have more anxiety about their access to treatment, which may contribute to their decision making about when they wish to be treated for their thyroid cancer, thus impacting thyroid cancer management. Our sample size limited further investigation of potential underlying mechanisms.

One major strength is that this study is among the first to analyze the impact of pre-existing anxiety and/or depression on patients with thyroid cancer. Similarly, this study is the first to identify a potential difference in time to obtain treatment for PTC based on pre-existing anxiety and/or depression. This study also highlights the importance of patient mental health in cancer care, a topic of growing focus within the literature.

## 5. Conclusions

Ultimately, our study represents the first of its kind to investigate the role of patient mental health on thyroid cancer management and decision making. We found a potential association between pre-existing anxiety and/or depression and a shorter interval between diagnosis and surgical intervention. Enhancing communication between patients and physicians and delivering patient-centered care by providing the oncology team with feedback about patients’ anxiety levels and depression may lead to improved care for patients with papillary thyroid cancer.

## Figures and Tables

**Table 1 cancers-16-00545-t001:** Characteristics of the study population by past diagnoses of anxiety, depression, or both anxiety and depression.

	No Anxiety or Depression (*n* = 560)	Anxiety (*n* = 29)	*p*-Value	Depression (*n* = 27)	*p*-Value	Both Anxiety and Depression (*n* = 36)	*p*-Value
Age—Mean (SD) (years)	51.8 (15.8)	53.2 (17.2)	0.488	52.9 (16.7)	0.74	54.1 (18.1)	0.163
Sex *n* (%)							
Male	190 (33.9)	<11	0.748	<11	0.097	13 (36.1)	0.79
Female	370 (66.1)	20 (69.0)		22 (81.5)		23 (63.9)	
Race *n* (%)							
Black	26 (4.6)	<11	<0.0001	<11	0.086	<11	0.003
White	274 (48.9)	24 (82.8)		20 (74.1)		28 (77.8)	
Not reported	29 (5.2)	<11		<11		<11	
Other	231 (41.3)	<11		<11		<11	
Insurance *n* (%)							
Private	325 (58.0)	18 (62.1)	0.006	12 (44.4)	0.238	19 (52.8)	0.022
Medicare	66 (11.8)	<11		<11		11 (30.6)	
Medicaid	115 (20.5)	<11		<11		<11	
Other/ unknown	54 (9.6)	<11		<11		<11	
Family history of TC *n* (%)							
Yes	55 (9.8)	<11	0.539	<11	0.321	<11	0.043
No	388 (69.3)	21 (72.4)		17 (63.0)		29 (80.6)	
Not reported	117 (20.9)	<11		<11		<11	
Comorbidities *n* (%)							
0	320 (57.1)	16 (55.2)	0.808	15 (55.6)	0.159	14 (38.9)	0.075
1	127 (22.7)	<11		<11		10 (27.8)	
2 or more	113 (20.2)	<11		<11		12 (33.3)	
Smoking *n* (%)							
Never	407 (72.7)	22 (75.9)	0.867	16 (59.3)	0.151	22 (61.1)	0.243
Former/current	135 (24.1)	<11		<11		13 (36.1)	
Not reported	18 (3.2)	<11		<11		<11	
BMI *n* (%) (kg/m^2^)							
<25	220 (39.3)	<11	0.05	<11	0.039	13 (36.1)	0.886
25–30	161 (28.8)	≥11		<11		<11	
>30	179 (32.0)	<11		15 (55.6)		13 (36.1)	
Unknown	<11	1 (3.5)		<11		<11	
Surgery type *n* (%)							
Total thyroidectomy	308 (55.0)	21 (72.4)	0.101	14 (51.9)	0.542	18 (50.0)	0.165
Subtotal thyroidectomy	33 (5.9)	<11		<11		<11	
Lobectomy w/isthmus	113 (20.2)	<11		<11		<11	
Lobectomy only	101 (18.0)	<11		<11		<11	
Isthmectomy	<11	<11		<11		<11	

**Table 2 cancers-16-00545-t002:** Time between papillary thyroid cancer diagnosis and surgery date.

	Mean (Days)	SD (Days)	*p*-Value	Median	IQR
No anxiety or depression	111.9	386.8		44	(28.5,83)
Anxiety	51.6	52.1	0.002	34	(56, 57)
Depression	57	48.3	0.004	50	(32, 63)
Anxiety and depression	57.4	47.3	0.003	42.5	(28.5, 76)
	β	*p*-value	β_adj_ *	*p*-value	
No anxiety or depression	REF		REF		
Anxiety	−60.31	0.402	−50.87	0.489	
Depression	−54.9	0.462	−37.76	0.633	
Anxiety and depression	−54.5	0.399	−51.29	0.443	

* adjusted for age, sex, race, insurance, and BMI.

## Data Availability

Data are contained within the article.

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
