# Peer review of "Anxiety and Depression as Potential Predictors for Shorter Time to Undergo Initial Surgical Treatment for Papillary Thyroid Cancer"

_cancers, 2024, doi:10.3390/cancers16030545_

Round 1

Reviewer 1 Report

Comments and Suggestions for Authors

I appreciate the opportunity to review this paper. My main expertise is in biostatistics and epidemiology. This paper is interesting in that is the first one I see linking this phenomenon of mood disorders and how they affect the time to surgical treatment. I believe this is a great idea that warrants further development.

The paper is written well although there are several obvious mistakes by the authors that need to be corrected. These are leaving the template instruction notes in lines 125-127 and 151-154 and by not providing in the paper able 3 even when it is mentioned in line 146. Please be more careful when proofreading before submission.

I have a concern with the participant selection as the people within the population that does not have a mood disorder may have the mood disorder but was just not reported. This is particularly problematic since the mood disorders are self-reported. How reliable is the predictor? How would this affect the outcome? What are the safeguards in the study to address this possible issue?

Please revise the mistakes and clarify the participant selection rationale and process.

Reviewer 2 Report

Comments and Suggestions for Authors

This paper describes the association of patient anxiety/depression on patients' decision to chose active surveillance over immediate surgery using time to surgery as the outcome. This association is, I believe, an important but rarely studied relationship. I am concerned about the method by which anxiety/depression were measured, and how these variables were analyzed, leading to the conclusion that greater patient anxiety/depression leads to faster time to treatment. Below are my specific comments-concerns. 

1. It is unclear if all patients were required to answer the same 2 (?) items asking about anxiety, and or depression. I would like to know how each of these items were worded, and if the response choice was no/yes or some other response. The authors state that the questions were self-report, but were the answers in response to a paper and pencil questionnaire-survey, are to an in-person interview? 

2. Give that there are multiple existing measures with known psychometric properties for assessing both (a) anxiety, and (b) depression, why did the authors chose to use only 2 single-item questions about anxiety and depression? Did the authors write these items themselves, or copy them from an existing measure? 

3. I am unable to ascertain if the authors are suggesting that both anxiety and depression are mood disorders? From my knowledge of clinical psychology, in the DSM-5-TR anxiety-related disorders are classified separately from mood disorders like depression. The description in the methods, results, and discussion treats these as nearly equivalent conditions, which I believe is erroneous. Also, the authors should  do a chi-square test or other statistic to report how closely associated the anxiety question is with the one on depression and inform the reader. 

4. The regression analyses performed remain unclear to me. I think that the outcome (time to surgery) should be regressed comparing non-anxious patients with (1) anxious patients, then (2) non-depressed patients comparing depressed patients, and (3) patients without either anxiety or depression on patients who have both conditions. This may involve using many of the same patients without anxiety or depression in more than 1 analysis, but may more clearly demonstrate any differences in the impact of anxiety vs. depression-alone, vs. anxiety+depression on the outcome. 

5. Related to #4, it seems to me that the effect of anxiety alone may be different than the effect of depression. More anxious patients may prefer to have relatively more radical treatment (i.e., surgery) vs. active surviellance and to receive this treatment as quickly as possible; whereas depressed patients may tend to feel more hopeless about their condition, and possibly feel less hurried or a greater desire to avoid treatment, leading to longer time to surgery. Without considering the potential opposing, or separate effects of anxiety vs. depression, the authors risk elucidating these potentially contradictory behaviors. 

6. I am unable to readily determine how many anxious/depressed patients opted for active surveillance vs. immediate surgery. Would this not be an interesting hypothesis to test as a separate question? Perhaps if only conducted as a 2x2 chi-square analysis? 

7. The authors note that type of insurance differs between "mood disorder" vs. no mood disorder patients. A greater percentage of mood disorder patients have Medicare (23.9%) compared with non-mood disorder patients (11.8%). The authors may want to examine whether this association is affected by patient age-group; or consider if patients on Medicare may have more anxiety or concern about their access to treatment? 

8. I think it is also noteworthy that patients with a family history of TC more frequently have "mood disorder" (15.2%) compared with non-mood disorder cases (9.8%). 

9. On line 119, it would be appropriate to describe the t-test as "Student's t-test." Note that "Student" is the published name of the author of the original publication describing the t-test.  

10. On line 296 in References, the date the BMI website was consulted appears incomplete, i.e., truncated. 

Comments on the Quality of English Language

The quality of the English in this manuscript is fine. 

Round 2

Reviewer 1 Report

Comments and Suggestions for Authors

My few concerns are satisfied with the authors clarification. The addition in the text that clarifies the self-reporting argument of the mood disorders is very appropriate.

I am happy to endorse the current draft for publication.